# Hyaluronic acid: Function and location in the urothelial barrier for bladder pain syndrome/interstitial cystitis, an in vitro study

**Charlotte J. van Ginkel** [ID]°*, **Cléo D.M. Baars**°, **Dorien M. Tiemessen** [ID],
**Cornelius F.J. Jansen, Frank M.J. Martens** [ID]**, Jack A. Schalken** [ID]**, Dick A.W. Janssen**

Department of Urology, Radboud University Medical Center, Nijmegen, The Netherlands

☯ These authors contributed equally to this work.
* charlottevanginkel@hotmail.com

## Abstract

Disruption of the glycosaminoglycan (GAG)-layer and urothelial barrier is an important aspect of the pathophysiology of bladder pain syndrome/ interstitial cystitis. Intravesical hyaluronic acid (HA) is often used in treatments for IC/BPS, however the role in the urothelial barrier is unknown. This study aims to clarify the location and functional contribution of HA in the urothelium, using an in vitro model. Immunohistochemistry was performed on human and porcine biopsies and on porcine cell cultures to evaluate the location of HA. Functional contribution was assessed through transepithelial electrical resistance measurements and the effects on gene expression in a differentiated primary porcine urothelial cell model. HA was found throughout in the urothelium and most abundant around the basal layer. Digestion of HA increased impermeability of the urothelium, contrasting with the effect of protamine sulfate (PS). After HA digestion, quantitative PCR analysis revealed upregulation of HA-synthesizing gene (HAS3) and the inflammatory marker (IL8). Treatment with HA and/or chondroitin sulfate therapy in undamaged cells upregulated genes related to GAG synthesis, barrier markers and inflammation. In PS-damaged cells, GAG therapy only upregulated genes associated with HA synthesis and inflammation, without affecting barrier recovery speed. These results emphasize the interaction of HA on urothelial cell inflammation and barrier repair physiology. HA seems to not directly restore the urothelial luminal GAG layer but influences barrier integrity through its interactions with urothelial cells.

## Introduction

Bladder pain syndrome/ interstitial cystitis (IC/BPS) is a poorly understood inflammatory condition of the bladder. The specific pathophysiology is unknown, but there is evident involvement of the immune system and a clear disruption of the urothelial barrier, making the bladder vulnerable to toxins and bacteria in the urine, leading to symptoms like pain and urinary frequency [1,2].

The urothelial barrier is predominantly formed by specialized tightly bound umbrella cells, covered by asymmetrical membrane units (consisting of uroplakins) and a layer of

**Data availability statement:** All relevant data are within the manuscript and its Supporting information files.

**Funding:** The research was financially sponsored by Goodlife Pharma. The funder did not play any role in study design, data collection and analysis, decision to publish or preparation of the manuscript.

**Competing interests:** Author Dick A.W. Janssen has received government support (ZonMW grant) for research regarding IC/BPS and GAG therapy. Authors have no further conflicts of interest to report.

hydrophilic polysaccharide components (e.g., glycosaminoglycans (GAGs)), also called the GAG-layer [1,2]. Specifically, this GAG-layer, normally a highly impermeable layer, is thought to be of great importance in the pathophysiology of IC/BPS. GAG-molecules associated with the urinary bladder are hyaluronic acid (HA), chondroitin sulfate (CS), heparan sulfate(HS) and dermatan sulfate(DS) [3–5]. The first two form the target for GAG-therapy bladder instillations, widely used in IC/BPS patients [6]. CS, HS and DS are sulfated GAGs, there synthesis involves attachment to a core protein, whereas HA is a non-sulfated GAG. CS has been found in the apical urothelium and has a significant role in the urothelial barrier. CS digestion with specific enzymes increases permeability and CS instillations help restore impermeability after damage [7,8].

The localization and function of HA in IC/BPS and urothelial barrier is less clear. It has been described in deeper layers of the urothelium [9,10]. HA is an unique non-sulfated GAG, a biopolymer with varying chain lengths, but with an overall high molecular weight (HMW) of approximately 1000 kDa in healthy native tissues [11,12]. HA synthesis occurs at the plasma membrane through the activity of specialized integral membrane glycosyltransferases known as hyaluronan synthases (HASs), there are three distinct HAS genes encoding the isoenzymes HAS1, HAS2, and HAS3. These enzymes coordinate the utilization of substrates to initiate, elongate and translocate HA, ultimately depositing it into the extracellular environment [13]. HA is found in many tissues, where in the ECM it exerts a role in tissue structure and cellular motility, adhesion and proliferation, along with interactions through the CD44 and RHAMM receptors [12,14–18]. The function of HA is closely linked to its molecular weight. In situations of tissue damage, HMW HA undergoes degradation, resulting in smaller fragments: low molecular weight (LMW) HA and/or oligo HA. These smaller HA particles have been associated with stimulation of inflammatory mediators, corresponding to an activated cell state, whereas HMW HA inhibits the production of these inflammatory mediators [19,20].

Despite HA being used in clinical treatment for IC/BPS and cystitis prevention, much is still unknown about its role(s) in the bladder wall. An in vitro IC/BPS model by Rooney et al., showed a decrease in inflammatory cytokines after HA therapy [21].

It is crucial to understand the localization and function of HA in healthy and inflamed urinary bladders. This study firstly aimed to give a novel insight in the location of HA in the normal urothelium and bladder wall and give new insights in various aspects of the functional contribution of HA in bladder wall homeostasis. For this we have developed a matured, terminally differentiated urothelial cell culture model with consistent high barrier properties. The first aim of this study was to validate epithelial barrier, inflammatory responses and recovery after damage in this primary urothelial cell culture model. We therefore assessed the degree of differentiation with immunohistochemistry (IHC) markers in the primary cell culture model and compared these with human and porcine bladder biopsies. Additionally, we performed functional assessments (transepithelial electrical resistance (TEER) measurements). Secondly, the localization of HA was assessed through additional IHC, the functional contribution of HA was analyzed through its effect on TEER and gene expression of genes coding for GAG synthesis, inflammation markers and barrier markers. The third aim of this study was to evaluate the functional effect of three clinically applied GAG therapies on an inflamed in vitro model.

## Materials and methods

### Materials, antibodies and other reagents

Unless specified otherwise, all chemicals were obtained from Merck (Merck KGaA, Darmstadt, Germany). The GAG formulations were chosen based on use in clinic and their

compound properties. The following GAG formulations were used; combined HA 1.6% (molecular weight 1400–2000 kDalton), sodium CS 2% (Ialuril®, IBSA), CS 2 mg/ml (Gepan instill®, Pohl-Boskamp GmbH&Co, Hohenlockstedt, Germany) and HA 1.6 mg/ml (molecular weight 2000 kDalton) (Instylan, Diaco biofarmaceutici, Trieste, Italy). Protamine sulfate (PS) (1400 IU/ml, 5 ml ampul) was obtained from LEO Pharma, Neu-Isenburg, Germany via the hospital pharmacy. Hyaluronidase was purchased from Sigma Aldrich (Saint Louis, United States, H3506 (specific activity 748 U/mg)).

## Cell culture

Urothelial cell isolation and culture was performed as described earlier by our research group [7,8]. In short, freshly dissected porcine bladders were obtained from a local abattoir and processed within 3 hours. In the morning, under sterile conditions porcine bladder specimens of 1 cm² were collected in transport medium (500 ml Hank's balanced salt solution (HBSS) completed with 10 mM HEPES, 100 U/ml penicillin and 100 µg/ml streptomycin and 1 µg/mL aprotinin). At the end of the day, the specimens were transferred and incubated in stripper medium overnight at 4 °C (HBSS, completed with 10 mM HEPES, 100 U/ml penicillin/ 100 µg/ml streptomycin and 2,4 U/ml Dispase II). The next day, isolation of urothelial cells was done by scraping the tissue and resuspending the cells in keratinocyte serum-free medium (K-SFM). The K-SFM was supplemented with 100 U/ml penicillin, 100 µg/ml streptomycin, 30 ng/ml cholera toxin, 50 µg/mL bovine pituitary extract and 5 ng/mL epidermal growth factor (Life Technologies Europe BV, Bleiswijk, Netherlands). The normal porcine urothelial (NPU) cells were seeded in T75 Primaria culture Flasks (Falcon, Corning, Glendale, United States) and placed at 37 °C in 5% v/v CO$_2$. The medium was refreshed three times per week. The cells were expanded after several passages, and passage three up to five were used for the experiments. Differentiation was induced in all settings by adding 5% foetal calf serum v/v and 2 mM Calcium chloride.

- For IHC: 50,000–75,000 cells were seeded in 8-well chamber slides (LabTek II; 15434), confluency was reached in approximately seven days. Followed by seven days of differentiation. The cells were fixated with 100% cold methanol.

- For TEER (Ω·cm²) measurements: 100,000 cells were seeded per insert in a 24-transwell plate, Ø 6.5 mm, pore size 0.4 µm (Costar, Fisher Scientific, Hampton, United States). Confluency was achieved in approximately one week, and differentiation was induced a few days later. After three weeks in total, the cells reached TEER values, resembling a very tight epithelium (2500–3500 Ω·cm²) [22–25]. For IHC the inserts were fixated in 3% paraformaldehyde and afterwards embedded in paraffin.

- For Gene Expression analysis: 100,000 cells were seeded per well in a 12-wells plate (Costar, Fisher Scientific, Hampton, United States). Confluency was reached in approximately a week, followed by one week of differentiation.

## Immunohistochemistry

The human bladder tissues were obtained from the Radboud Biobank Urology, a repository managed by the Department of Urology, which includes donated blood and other biological materials. Three human tissues were accessed on 28-10-2021, 14-10-2022 and 18-08-2022. There was no access to information that could identify participants regarding these tissues. Three porcine biopsies were used as well. Frozen porcine and human bladder tissues were cut into 4 µm sections and fixed using 3% paraformaldehyde in 10 minutes.

Cell cultures were processed as described above. Heat induced epitope retrieval was performed with 10 mM Sodium Citrate buffer (6.0) for 10 minutes. The primary antibodies were used: zonula occludens-1 (tight junctions) (Invitrogen, Thermo Fisher Scientific, Waltham, United States), RGE53 (cytokeratin 18) (Nordic Mubio, Susteren, Netherlands), HA receptors CD44 (Thermofisher Scientific) and RHAMM/CD168 (ITK diagnostics). Information regarding antibodies can be found in Table 1. Conjugate Goat-a-Mo. Rab/Rt-HRPO (Immunologic B.V., Arnhem, Netherlands) was used as the secondary antibody. Bound antibodies were visualized by DAB (Immunologic B.V., Arnhem, Netherlands) and hematoxylin counterstain.

For staining of HA, the biotinylated hyaluronan binding protein (HABP) was used. For negative control hyaluronidase 4 mg/ml in 10 mM phosphate buffered saline (PBS) was incubated for 1 h at 37 °C. The antigen retrieval was followed by blocking with avidin/biotin (Vector) and normal goat serum (Bodinco BV, Alkmaar, Netherlands). HABP (1:400) was applied for 1 h at room temperature, followed by streptavidin Poly HRP (Pierce, Thermo Scientific, Hampton, United States). Visualization was achieved with DAB and hematoxylin counterstain as well. Bright field microscopy (Zeiss, Stuttgart, Germany) was used for analysis.

## Barrier function model

The epithelial barrier was assessed by real time TEER measurements, that allows electrical resistance to be measured across a cultured layer of epithelial cells, without damaging the cells or cellular barrier function. For this, the Millicell ERS-2 Voltohmmeter (Merck KGaA, Darmstadt, Germany) was used to measure TEER on differentiated NPU cells grown in a tight epithelial layer on a semi permeable polycarbonate membrane in a transwell insert. In short, the contribution of HA was investigated by applying hyaluronidase 4 mg/ml in K-SFM for 24 h at 37 °C in 5% v/v $CO_2$ on the apical side of the cell layer. PS treatment consisted of apical application of 1400 IU/ml for 3 h at 37 °C in 5% v/v $CO_2$, removed afterwards and washed three times with 0.9% sodium chloride. Untreated samples, used as controls, were also washed three times with 0.9% sodium chloride. TEER was measured in hours; at baseline T = 0, T = 3, T = 5, T = 7, T = 12 and T = 24. The untreated samples were kept in culture and followed up for an extent of in total 55 days. Each group consisted out of eight replicates. Experiments were performed in duplo.

For the effect of GAG therapy on the recovery of TEER after damage, cells were pretreated apically with PS, 1400 IU/ml for 3 h, followed by one of the three GAG therapies added apically (16 mg/ml HA & 20 mg/ml CS, 2 mg/ml CS alone and 1.6 mg/ml HA alone) for 1 h. Untreated cells (negative control) and PS alone treated cells served as controls. In between

**Table 1. Antibodies used for IHC.**

|  | HAPB[1] | ZO-1 | RGE53 | CD44 | RHAMM/CD168 |
|---|---|---|---|---|---|
| Monoclonal/polyclonal | – | Monoclonal | Monoclonal | Monoclonal | Monoclonal |
| Host species | – | Mouse | Mouse | Mouse | Rabbit |
| Catalogue nr. | 385911 | 33–9100 | MUB0326P | BMS113 | Ab108339 |
| Antigen | – | ZO-1 | IgG1 | Human CD44std | CD168 antibody |
| Dilution | 1:400 | 1:200 | 1:200 | 1:40.000 | 1:100 |
| Stable public identifier | – | AB_2533147 | – | AB_10597135 | AB_10861654 |
| Supplier | Millipore | ThermoFisher Scientific | Nordic MUbio | ThermoFisher Scientific | ITK diagnostics |

[1]Biotinylated Hyaluronic Acid Binding Protein (HABP) is not an antibody.

treatments all cells (including controls) were washed three times with 0.9% sodium chloride. TEER was measured in hours; at baseline T = 0, T = 3, T = 5, T = 7, T = 12 and T = 24. Each group consisted out of eight replicates.

## Gene expression

The genes analyzed are summarized in Table 2. This includes the following GAG synthesizing genes (*HAS2, HAS3, CSGALNACT1, CSGALNACT2, HSPG2, SDC1*), inflammatory markers (*IL8* and *IL6*) and barrier markers (*TJP1, OCLN, UPK3A, CDH1*). The same experimental setup employed for the barrier function experiments was utilized. In short, this included the following groups: 1) hyaluronidase treatment 4 mg/ml, 2) PS treatment 1400 IU/ml, 3) PS pre-treatment, followed by 16 mg/ml HA & 20 mg/ml CS therapy, 4) PS pre-treatment, followed by 2 mg/ml CS therapy, 5) PS pre-treatment, followed by 1.6 mg/ml HA therapy. Untreated samples were used as controls. Cells were harvested at the following timepoints in hours; at baseline T = 0, T = 3, T = 5, T = 7, T = 12 and T = 24. In addition, an experiment involving undamaged cells was conducted. These cells received one of the three GAG therapies (without pretreatment of PS) for 1 hour and were subsequently washed 3 times with 0.9% sodium chloride and harvested 1 hour later (T = 2). The experiment was performed twice.

RNA was isolated with TRIzol (Life Technologies ThermoFisher Scientific, United States) and quantity assessed by nanodrop (ND-2000 Spectrophotometer (ThermoFisher Scientific, United Kingdom). RNA was stored at −20 °C. cDNA synthesis was performed using hexamer primers and SuperScript II RT (Thermo Fisher Scientific, Waltham, USA). All cDNA was stored at −20 °C. SYBR Green qPCR was performed using gene specific primers and a Light-Cycler 480 machine, according to manufacturer's instructions (Roche, Basel, Switzerland). All reactions were performed in duplo. Relative expression was assessed by normalization to housekeeping gene *GAPDH* using the ΔΔ CT method.

## Statistical analysis

All experiments were performed in duplo. Treatment groups were compared using the ANOVA test and a Bonferroni post-hoc analysis (normally distribution) and Kruskal Wallis

**Table 2. Gene primer sequences primers for qPCR.**

| Gene | Forward primer | Reverse primer |
|---|---|---|
| Ssc CSGALNACT1 | CTGCGACGTGGACATCTACT | ACTGACTGAAAAGGACCGGAT |
| Ssc-CSGALNACT2 | GATGAAGATGGTCCCCTTGGA | GTGCCCTTATCTCTCTCCGTG |
| Ssc-HAS1 | TGTGTCCTGCATCAGTGGTC | TCCCCAAAGGTACAGTGGGT |
| Ssc-HAS2 | CAGACAGGCTGAGGACAACT | CCAAGGAGGAGAGAAACTCCAA |
| Ssc-HAS3 | CCTTGGTGACTCCGTGGAC | AGCATCTCGAAGGTACAGGC |
| Ssc-HSPG2 | CTGCCCATTGTACAGCAGC | CCCACCATCAAGGATGCCTAC |
| Ssc-SDC1 | CGTTTTGGCTCTGGCTCTGC | ACATTGTGGCCACGGTTTC |
| Ssc-IL6 | TGGCAGAAAAAGACGGATGC | CCACAAGACCGGTGGTGATT |
| Ssc-IL8 | CTGTTGCCTTCTTGGCAGTTT | TGCACTTACTCTTGCCAGAACT |
| Ssc-OCLN | CGTCAGGTGCACCCTCCA | TGGACTTTCAAGAGGCCTGG |
| Ssc-TJP1/ZO-1 | GGCCCCATGGTCTCAAGTTC | AAGACACTTGTTTTGCCAGGT |
| Ssc-CDH1 | TGGGAGGCATCCTTGCTTTT | ACCACCCTTCTCCTCCGAAT |
| Ssc-UPK3A | TCGTTATCACGTCCATCCTG | TGTATGAAGGCTCCGAGGTC |

test (not normally distributed). Data are presented as mean ± standard deviation, statistical significance indicated as * $p < 0.05$, #$p < 0.05$, **$p < 0.001$ and ##$p < 0.001$.

## Results

### Characterization of HA, its receptor and the tight junction network

Umbrella cells were seen apically in the human- and porcine bladder biopsies and in the NPU culture (Fig 1). Through staining of cytokeratin 18 their typical morphology was clearly visualized. Secondly, a well-developed network of tight junctions between umbrella cells was shown though ZO-1 staining in Fig 1.

Focusing on HA for human and porcine biopsies, HA was seen apical, but evidently most abundantly present around the level of the basal cell layer of the urothelium (Fig 2A.1 and A.3), mirroring the distribution of the CD44 receptor (Fig 2C.). HA was also present in the NPU cultures, shown in the most apically situated cells (Fig 2A.5) and the membranes of the cells just below (Fig 2B.1). In all biopsies and cell cultures, a proper digestion of HA by hyaluronidase was seen (Fig 2A.2, A.4, A.6 and B.2).

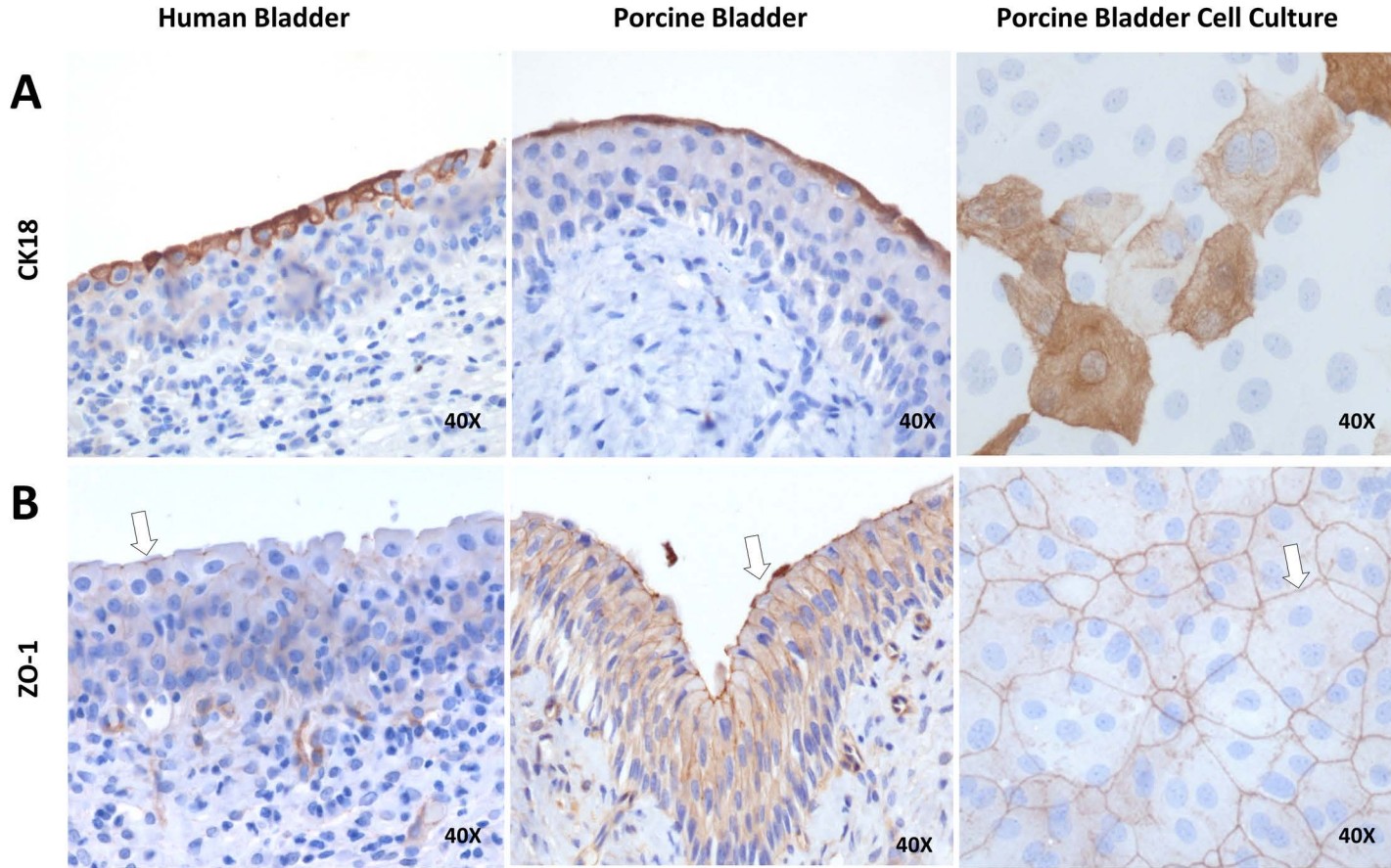

**Fig 1. Expression of umbrella cells and tight junctions in human- and porcine bladder biopsies and porcine urothelial cell culture.** Images of biopsied tissue of human bladder, porcine bladder and porcine bladder cell culture. (A) Umbrella cells are visualized in the most apical layer of human and porcine bladder biopsies and porcine bladder cell cultures (CK18). (B) The tight junctions (ZO-1) are seen as a thin belt apically in human and porcine bladder biopsies, in porcine cell cultures it is seen as a network over the urothelial cells (arrows).

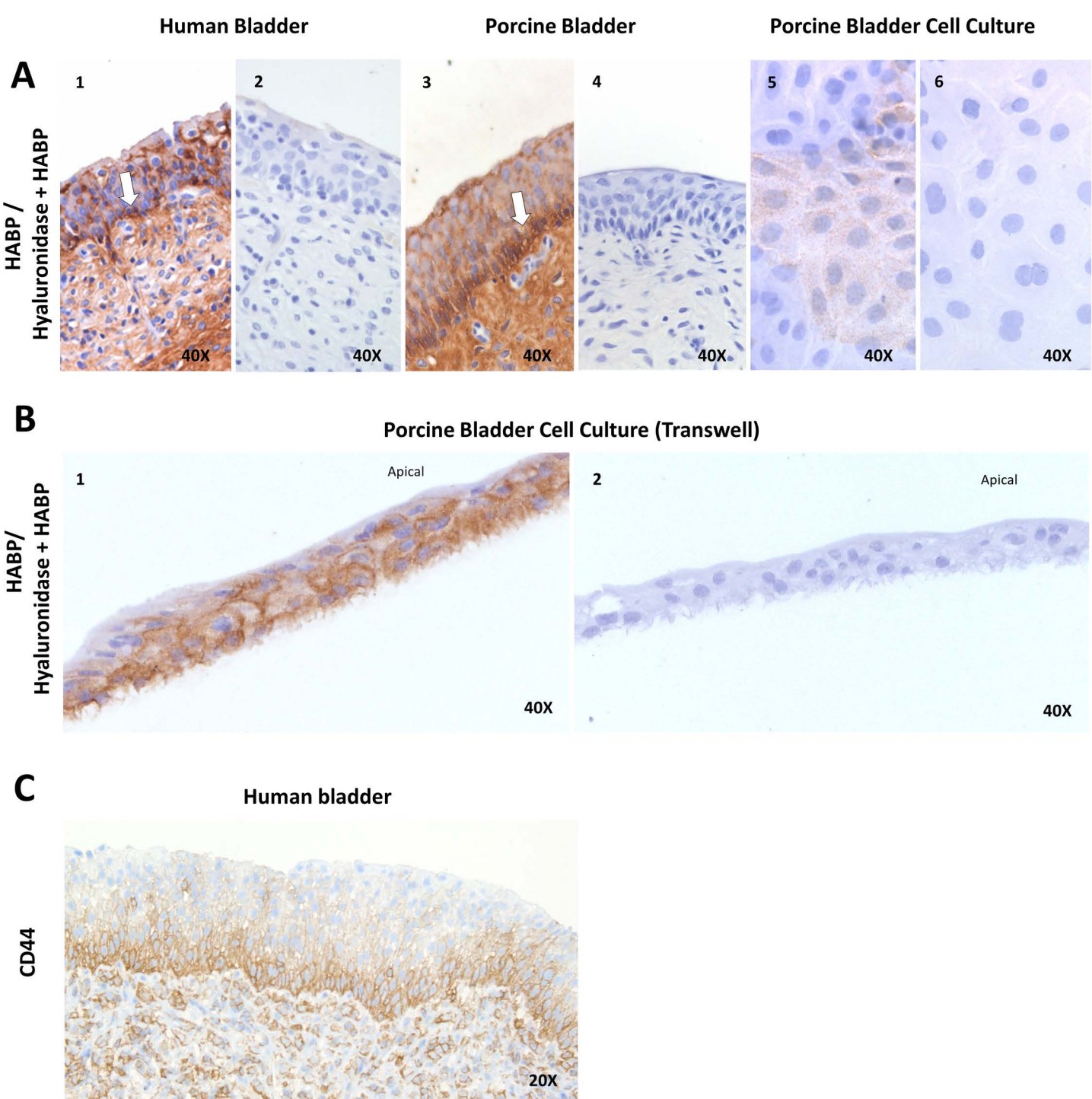

**Fig 2. Expression of HA and its receptor in urothelium.** Images of biopsied tissue of human bladder, porcine bladder and porcine bladder cell culture. Staining with HABP, digestion of HA was achieved with hyaluronidase (A2, A4, A6, B2). (A1, A3) HA is most pronounced expressed around the basal layer of the urothelium in human and porcine bladder biopsies (arrows). (A5, B1) HA expression apically and in cell membranes in primary porcine urothelial cell cultures. (C) The CD44-receptor is found around the level of the basal cell layer. HABP, hyaluronic acid binding protein; HA, hyaluronic acid.

## The effect of HA digestion and -replenishment alone or with CS on barrier function

The TEER of the NPU culture progressively increased over time, reaching and retaining a stable value of >3300 $\Omega \cdot cm^2$ (upper reliable limit of the equipment) around day 35 and a value of >4300 $\Omega \cdot cm^2$ (absolute upper limit of the equipment) around day 47 (Fig 3A). PS created a TEER drop of 44% that normalized after 24 hrs.

The effect of hyaluronidase treatment on the TEER value is illustrated in Fig 3B. Following hyaluronidase treatment, the initial TEER value of 2879 ± 306 $\Omega \cdot cm^2$ gradually increased to 3832 ± 302 $\Omega \cdot cm^2$ after 24 hours. This change in TEER was significantly different (p < 0.001) from both the untreated group, which stayed stable from 2950 ± 166 $\Omega \cdot cm^2$ to 3008 ± 350 $\Omega \cdot cm^2$ and the PS treated group (T = 0 3428 ± 323 $\Omega \cdot cm^2$), which initially decreased but recovered to 2962 ± 287 $\Omega \cdot cm^2$ in 24 h. No beneficial effect on the recovery of TEER was shown for the three GAG-therapies in damaged (PS treated) cells in the in vitro model (Fig 3C).

## Gene expression level

### Effect of digestion of HA on HA-synthesis and barrier- and inflammation markers.
NPU cells treated with hyaluronidase or PS showed a fast temporary increase in

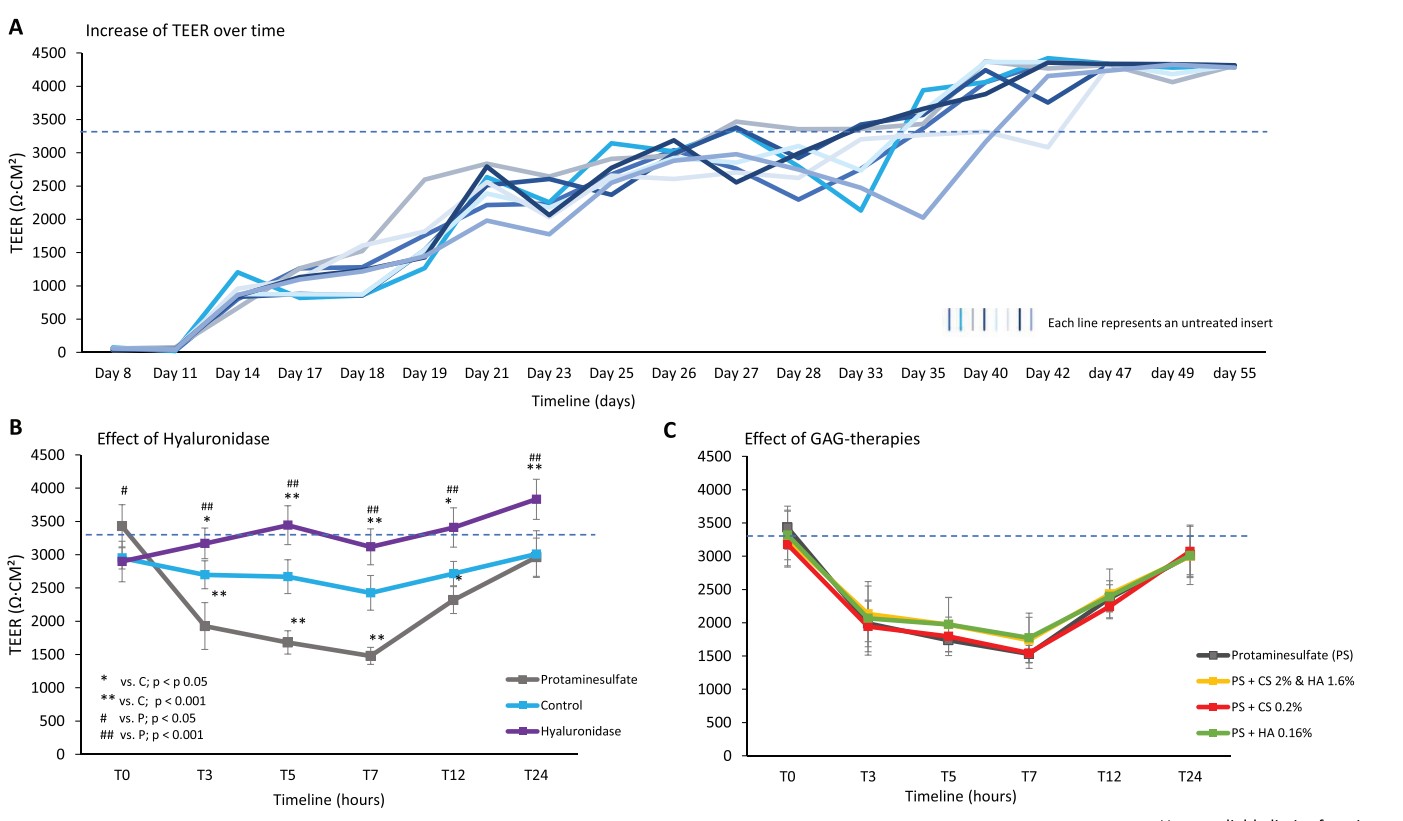

**Fig 3. The effect of HA digestion and HA replenishment alone or with CS on barrier function measured by TEER.** (A) A steady increase in TEER is seen in differentiated porcine urothelial cells, corresponding with a tight epithelium and reaching the upper reliable limit of the equipment (3300 $\Omega$ cm²) around day 35. In (B) and (C) each point represents a mean ± standard deviation. (B) Hyaluronidase treatment (n = 8) gradually increased TEER, this significantly differed from the untreated group (n = 8) and the PS treated group (n = 8) (p < 0,001). PS decreased TEER. (C) Different treatments (treatment groups n = 8)with HA and/or CS did not affect TEER recovery after PS treatment, full recovery was seen in all groups after 24 hours. TEER, transepithelial electrical resistance; PS, protaminesulfate; HA, hyaluronic acid; CS, chondroitin sulfate; T0, baseline; T3, 3 hours; T5, 5 hours; T7, 7 hours; T12, 12 hours; T24, 24 hours.

the expression of *HAS3*, reaching relative expression of >3x and >2x respectively (Fig 4). Moreover, for PS an increase in relative *HAS2* expression (>6x) was seen as well, but with low absolute expression. In NPU cells no expression of *HAS1* was found (data not shown). There was no clear and consistent effect of either treatment on the expression of other GAG and proteoglycan synthesizing genes (*CSGALNACT2, CSGALNACT1, HSP1, SDC1*).

Regarding inflammation markers, PS treated cells showed increased *IL8* and *IL6* expression. For hyaluronidase treated cells, this increase was only seen for *IL8*. There was no evident effect of either treatment on genes for tight junction proteins, uroplakin and E-cadherin, the small increase at 1 timepoint in *TJP1* and *OCLN* at T7 was not consistent over the experiments and therefore not considered specific.

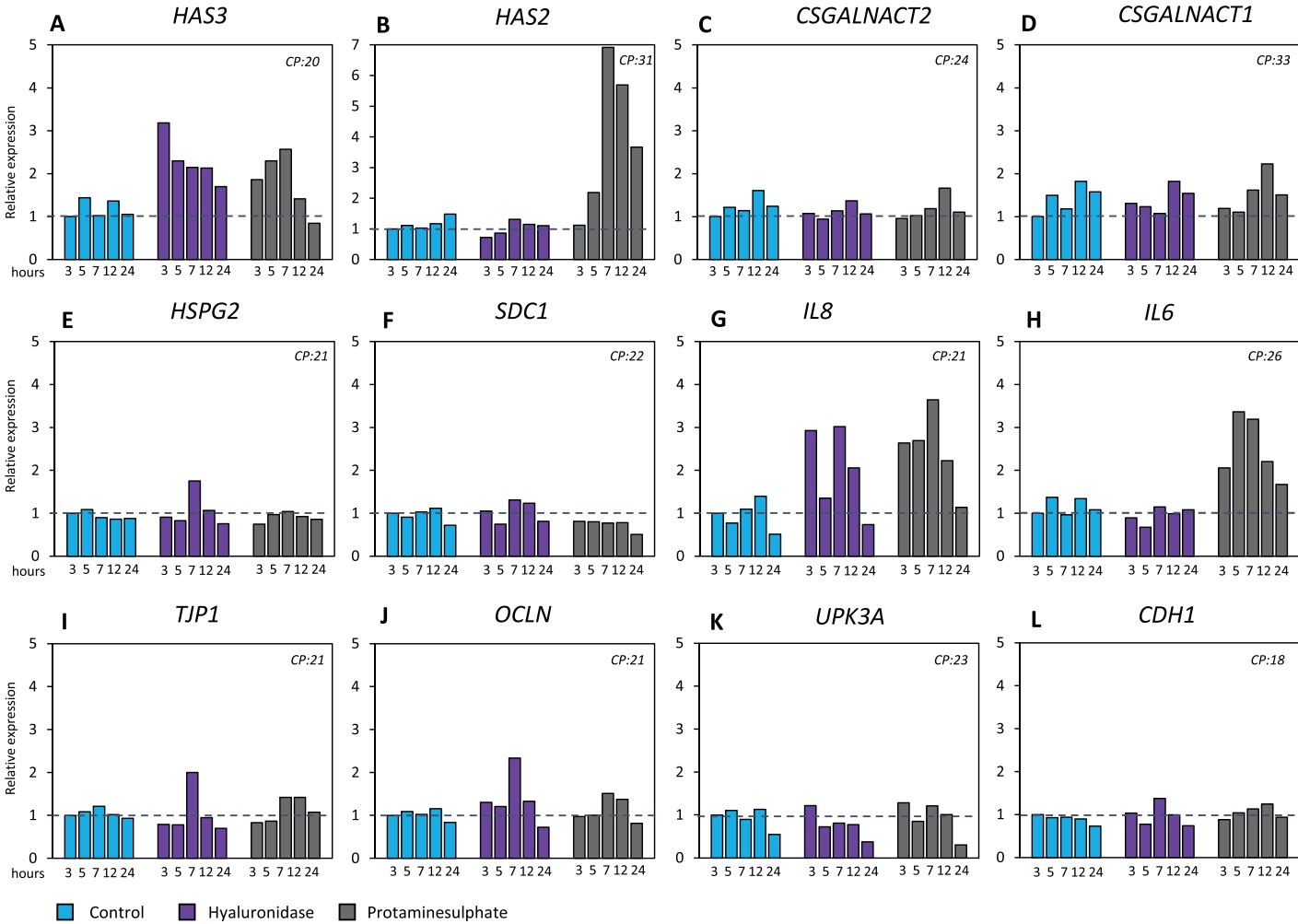

**Fig 4. The effect of HA digestion on gene expression level of HA synthesizing genes, other GAG synthesizing genes and barrier markers.** Effect of hyaluronidase treatment compared to PS treatment and untreated samples on gene expression of HA synthesizing genes, other GAG and proteoglycan synthesizing genes, inflammation markers and barrier markers. Expression of genes in hyaluronidase treated and PS treated samples are relative and compared to the untreated samples of that respective timepoint, which is y1 and represents the control line. (A, B) Hyaluronidase treatment increased relative expression of *HAS3*. Treatment with PS increased relative expression of *HAS3* and *HAS2*. (C, D, E, F) No effect was seen of treatment with hyaluronidase or PS on the expression of other GAG/proteoglycan synthesizing genes (*CSGALNACT2, CSGALNACT1, HSPG2, SDC1*) (G, F) Hyaluronidase increased the relative expression of *IL8*, PS increased expression of *IL8* and *IL6*. (I, J, K, L) No effect was seen of PS or Hyaluronidase treatment on barrier markers (*TJP1, OCLN, UPK3A, CDH1*). PS, protaminesulfate; T3, 3 hours; T5, 5 hours; T7, 7 hours; T12, 12 hours; T24, 24 hours; CP, median crossing point.

## The effect of replenishment of HA alone or with CS on HA synthesis, barrier- and inflammation markers

In undamaged cells treatment with HA&CS, CS, and HA led to an increase in the expression of all GAG synthesizing genes (*HAS3, CSGALNACT2, CSGALNACT1, HSPG2, SDC1*), except for *HAS2* where only treatment with HA increased the expression (Fig 5). The gene expression

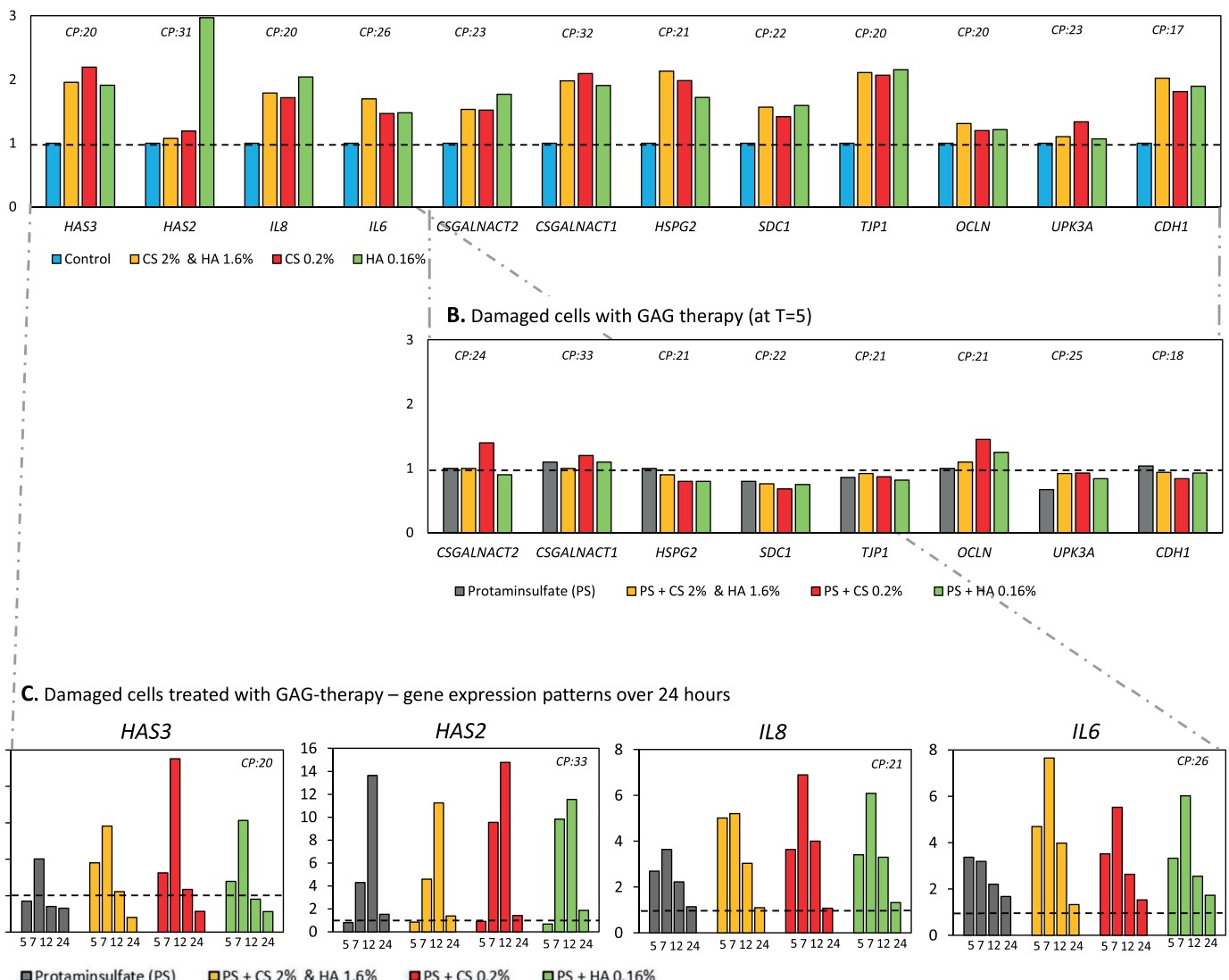

**Fig 5. The Effect of clinically applied GAG therapies after cell damage on gene expression level of HA and other GAGs synthesizing genes and inflammation and barrier markers.** Effect of clinically applied GAG therapies on gene expression of HA synthesizing genes, other GAG and proteoglycan synthesizing genes and inflammation and barrier markers. The evaluated GAG therapies are combined HA 1.6% with CS 2%, CS 0.2% and HA 0.16%. The expression is shown relatively to the untreated samples, which are represented by the dotted line. (A) Treatment with all GAG therapies increased the relative expression of almost all GAG producing genes *(HAS3, CSGALNACT2, CSGALNACT1, HSPG2, SDC1), HAS2* was only increased after isolated HA treatment. *IL6* and *IL8* as well as *TJP1* and *CDH1* increased in relative expression after all GAG therapies. (B) In damaged cells (pre-treated with PS) other GAG/proteoglycan synthesizing genes (*CSGALNACT2, CSGALNACT1, HSPG2, SDC1)* and barrier markers (*TJP1, OCLN, UPK3A* and *CDH1*) were not affected by treatment. (C) For *HAS3, HAS2, IL8* and *IL6* in damaged cells all GAG therapies lead to an increase. GAG, glycosaminoglycans; PS, protaminesulfate; HA, hyaluronic acid; CS, chondroitin sulfate; T5, 5 hours; T7, 7 hours; T12, 12 hours; T24, 24 hours; CP, median crossing point.

of the inflammatory cytokines IL-8 and IL-6 was also increased, together with genes coding for tight- and adherence junctions' formation.

PS damaged cells treated with three different GAG therapies showed rapid and relatively higher increase in expression of *HAS3* and *HAS2,* and inflammatory markers *IL8* and *IL6* compared to untreated cells. These increases were rapid and normalized within 24 hours. However, no additional effects were seen on the synthesizing genes of other GAG's, proteoglycans and barrier markers such as tight junctions and adherence junctions in PS damaged cells treated with GAG therapy.

## Discussion

In this study, the main objective was to clarify the precise localization and functional role of HA within the urothelium of the bladder. To comprehensively address these aims, it was crucial to ensure that the in vitro model accurately replicated a functional urothelium, characterized by functional differentiation of epithelial cells and impermeability, with the formation of umbrella cell, barrier markers such as tight junctions and high TEER values. The TEER values achieved with this in vitro model were extremely high (more than 4 times higher compared to previous research). [7,8] Cells remained viable with these conditions for over 55 days. Immunohistochemical analysis revealed a close resemblance between human and porcine bladders, a resemblance also evident in NPU cultures and shown before [7,23]. Corresponding with the observations of Turner et al., NPU cultures exhibited tight junctions and umbrella cells, even before the differentiation protocol was fully implemented. Tight junctions and umbrella cells, seen in histological IHC analyses, indicate the presence of differentiated cells and barrier markers. Our results show that this does not confirm impermeability, since these conditions still generated low TEER values (approximately < 200 $\Omega \cdot cm^2$) indicating a leaky epithelium [22,23]. This emphasizes the necessity of incorporating functional assessments such as TEER measurements for epithelial barrier research using in vitro models. After adding differentiation medium, our model successfully emulated an impermeable urothelium, with the properties essential for investigating our research inquires.

### HA location and function

HA was found most abundant in the deeper layers of the human and porcine urothelium, this is in line with earlier studies [9,10]. The greater presence of HA around the cell membranes (Fig 2B1) might be attributed to the HA production at the plasma membrane. Correspondingly, the HA receptor CD44 was identified around the basal cell layer of the urothelium, consistent with prior findings as well [26]. Comparatively, the GAG composition in the deeper urothelial layer differs from that in the apical layer, of which the latter is mostly composed of CS [7,10,27]. Our results did not find a direct involvement of HA as a barrier increasing GAG. Enzymatic digestion of HA in cell cultures did not lead to an increase in permeability, but unexpectantly resulted in a more impermeable barrier compared to untreated cells. These findings are contrary to the effects seen during the removal of CS [7]. When considering the location of CS and HA within or on top of the urothelium, it appears that the luminal presence of CS directly influences the barrier integrity and the GAGs found deeper within the urothelium, such as HA, do not seem to directly compromise the barrier following digestion [8]. Similar findings were previously described for the basally located heparan sulfate [7]. As to the mechanism of increase in impermeability after enzymatic HA digestion, we can currently only speculate. There is literature that describes a mediating role for the TSG6 protein that interacts between the HA and the CD44 receptor that is worthwhile exploring for future research to further clarify this. On a cellular level, HA function and composition determined

by different chain lengths are mediated via hyaluronidase and mediates cellular metabolism, cell surface proteins, cellular adhesion and consequently cell proliferation and migration. Even the HA breakdown products are biologically active and activate Toll-like receptors that activate inflammatory responses [28].

In response to inflammation (PS exposure), there was a robust and rapid increase in the expression of *HAS2* and *HAS3*. The rapid dynamic response of HA has been suggested to be possibly due to the simplicity of HA synthesis. HA production is regulated without affecting other GAG production in the Golgi, a potential explanation is the more complex regulation of the other GAGs, which involves not only controlling the GAG synthetic machinery but also managing the expression of core proteins [29]. Rapid *HAS2* and *HAS3* upregulation is in correspondence with other studies looking into ketamine induced cystitis [19].

Removal of HA led to positive feedback through predominantly one HA synthesizing gene, *HAS3*. Though the HAS genes are similar in function and protein sequence, *HAS3* codes in particular for LMW HA, while *HAS2* is thought to result more in HMW HA, [30,31]. Hyaluronidase cleaves the HA polymer [32] potentially yielding LMW HA and/or oligo HA. Contrary to HMW HA, LMW HA has been shown to promote the production of proinflammatory cytokines [19,20]. The increased *IL8* expression could possibly be explained, by the increased amount of LMW HA due to cleaving of HMW HA and increased *HAS3* expression. The upregulation of other GAG, proteoglycan and tight junctions RNA after GAG therapy with HA alone or in combination with CS does imply a role for HA in barrier regulation of the bladder urothelium. The only contrasting finding in this study was that this effect was not seen in PS damaged conditions as one would expect. Our study results therefore demonstrate that downstream effect of HA suppletion on urothelial cells are complex and currently not fully understood. An inhibited or delayed response may be caused by the induced inflammatory trigger and subsequent mediation of HA receptors like CD44 and RHAMM. Also, the additional effects of HA breakdown products on inflammatory responses via the Toll-like receptors play a role, these factors should be further explored [28].

## HA replenishment therapy alone or with CS

The second objective of this study was to investigate the effect of three different GAG therapies on the recovery of damaged NPU cells. We did not find a beneficial effect on TEER recovery of the three GAG therapies. Contrary to earlier findings of our research group that demonstrated a significant beneficial effect of CS treatment [8]. This difference could be potentially explained because the current model results in a notably tighter baseline epithelium with higher TEER values, even after PS treatment. Suggesting that the cells were less susceptible to damage and more resilient in repair. The recovery in TEER over 24 hours in all four groups was in line with earlier findings [7,8].

Additionally, the effect of the GAG therapies was investigated on gene expression level. To do this comprehensively, first the effect of the therapies on healthy NPU cells was investigated (at T = 2). This showed an increase in the expression for the synthesis of HA, CS and HS. There was also an increase in expression for tight- and adherence junctions. These results suggest a fast response of the urothelium to generate a stronger urothelial barrier under influence of GAG therapies. The additional increase of pro-inflammatory cytokines under these circumstances could possibly be explained, as the acute phase reaction which sets off repair. This mild inflammatory response after GAG therapy was also seen in Rooney et al 2020 [33].

Moreover, in the damaged setting (after treatment with PS) the synthesis of HA is stimulated and even further upregulated when GAG therapies are given. In 2020 Rooney et al. also showed modulation of the HA receptor genes after GAG therapy with HA [33]. For the

other genes involved in GAG synthesis no evident modulation was seen, in contrary to earlier findings by Rooney et al. [21]. However, there are notable important differences between the applied models, including variations in cellular background, HA therapy dosage and duration. The application of well characterized differentiated primary cells and clinically relevant HA therapy dosage and duration in the applied in vitro model, along with the addition of functional assessments, makes the current model especially suited to adequately evaluate urothelial barrier properties.

Interestingly, inducing damage did not alter the expression of genes associated with barrier function as detected in undamaged cells. The additional therapy with GAGs also did not stimulate the expression of barrier molecules (also not after 24 hours, data not shown) as was seen in the undamaged setting. HA breakdown products or other downstream effects of protamine sulfate may sustain an inflammatory response and explain this impaired barrier recovery, but this needs further clarification [28]. Even so, the functional barrier recovered within 24 hours in all groups, possibly because of the regulation of other genes related to barrier properties. It is unclear if this lack of effect in the damaged model is due to some prolonged effects of PS or if other types of cell damage may yield a different response.

Concerning inflammatory cytokines, there is an increased expression of *IL8* and *IL6* after GAG therapy in damaged cells. High levels of IL-6 are associated with a high degree of complains of IC/BPS [34], additionally in vivo studies show lower IL-6 measurements in damaged bladders after weeks of GAG therapy [35]. Our results would seem contrary, but IL-6 is involved in the acute phase reaction and plays an important role in the timely resolution of wound healing. The signaling of IL-6 is responsible for the switch to a reparative environment [36–38]. This fits with our increased *IL6* (together with *IL8*) expression directly after inflammation, followed by normalization of the gene expression and full recovery of barrier function within 24 hours.

## Conclusion

Our results emphasize the biological interactions that HA has on urothelial cell inflammation and barrier repair physiology. Intravesical GAG therapies with HA does not seem to directly repair the urothelial luminal GAG layer but modulates this action through urothelial cell interactions. There is a need for future research to further explore possible very fast-acting mechanisms around HA and other GAGs, as well as the later effects concerning inflammation and repair. This is pivotal to fully clarify the kinetics of the urothelial response to damage and optimize potential therapeutic interventions.

## Supporting information

**S1 Data. Dataset PLOSONE.**
(XLSX)

## Acknowledgements

There were no additional contributors to the research or manuscript who are not listed as authors.

## Author contributions

**Conceptualization:** Charlotte van Ginkel, Cléo D.M. Baars, Dorien M. Tiemessen, Cornelius F.J. Jansen, Frank M.J. Martens, Jack A. Schalken, Dick A.W. Janssen.

**Data curation:** Charlotte van Ginkel, Cléo D.M. Baars, Dorien M. Tiemessen, Cornelius F.J. Jansen, Jack A. Schalken, Dick A.W. Janssen.

**Formal analysis:** Charlotte van Ginkel, Cléo D.M. Baars, Cornelius F.J. Jansen, Dick A.W. Janssen.

**Funding acquisition:** Dick A.W. Janssen.

**Investigation:** Charlotte van Ginkel, Cléo D.M. Baars, Dorien M. Tiemessen, Jack A. Schalken, Dick A.W. Janssen.

**Methodology:** Charlotte van Ginkel, Cléo D.M. Baars.

**Project administration:** Dick A.W. Janssen.

**Resources:** Jack A. Schalken, Dick A.W. Janssen.

**Supervision:** Frank M.J. Martens, Jack A. Schalken, Dick A.W. Janssen.

**Validation:** Dick A.W. Janssen.

**Visualization:** Charlotte van Ginkel, Cléo D.M. Baars.

**Writing – original draft:** Charlotte van Ginkel, Cléo D.M. Baars.

**Writing – review & editing:** Dorien M. Tiemessen, Cornelius F.J. Jansen, Frank M.J. Martens, Jack A. Schalken, Dick A.W. Janssen.

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
