## [Decision Letter · Decision Letter 0]

31 Oct 2024

PONE-D-24-21268Hyaluronic acid: function and location in the urothelial barrier for interstitial cystitis/ bladder pain syndrome, an in vitro studyPLOS ONE

Dear Dr. van Ginkel,

Thank you for submitting your manuscript to PLOS ONE. After careful consideration, we feel that it has merit but does not fully meet PLOS ONE’s publication criteria as it currently stands. Therefore, we invite you to submit a revised version of the manuscript that addresses the points raised during the review process.

**Thank you for submitting the following manuscript to PLOS ONE.**

**Please revise the manuscript according to the reviewers' comments and upload the revised file.**

We look forward to receiving your revised manuscript.

Kind regards,

Yung-Hsiang Chen, Ph.D.

Academic Editor

PLOS ONE

Journal Requirements:

2. Thank you for submitting the above manuscript to PLOS ONE. During our internal evaluation of the manuscript, we found significant text overlap between your submission and previous work in the [introduction, conclusion, etc.].

Please revise the manuscript to rephrase the duplicated text, cite your sources, and provide details as to how the current manuscript advances on previous work. Please note that further consideration is dependent on the submission of a manuscript that addresses these concerns about the overlap in text with published work.

[If the overlap is with the authors’ own works: Moreover, upon submission, authors must confirm that the manuscript, or any related manuscript, is not currently under consideration or accepted elsewhere. If related work has been submitted to PLOS ONE or elsewhere, authors must include a copy with the submitted article. Reviewers will be asked to comment on the overlap between related submissions (http://journals.plos.org/plosone/s/submission-guidelines#loc-related-manuscripts).]

We will carefully review your manuscript upon resubmission and further consideration of the manuscript is dependent on the text overlap being addressed in full. Please ensure that your revision is thorough as failure to address the concerns to our satisfaction may result in your submission not being considered further.

“author Dick A.W. Janssen has received government support (ZonMW grant) for research regarding IC/BPS and GAG therapy. Authors have no further conflicts of interest to report.”

Additional Editor Comments:

Thank you for submitting the following manuscript to PLOS ONE.

Please revise the manuscript according to the reviewers' comments and upload the revised file.

Reviewers' comments:

Reviewer's Responses to Questions

**Comments to the Author**

1. Is the manuscript technically sound, and do the data support the conclusions?

Reviewer #1: Yes

Reviewer #2: Yes

Reviewer #3: Partly

Reviewer #4: Yes

2. Has the statistical analysis been performed appropriately and rigorously? 

Reviewer #1: I Don't Know

Reviewer #2: Yes

Reviewer #3: Yes

Reviewer #4: I Don't Know

3. Have the authors made all data underlying the findings in their manuscript fully available?

Reviewer #1: Yes

Reviewer #2: Yes

Reviewer #3: No

Reviewer #4: Yes

4. Is the manuscript presented in an intelligible fashion and written in standard English?

Reviewer #1: Yes

Reviewer #2: Yes

Reviewer #3: Yes

Reviewer #4: Yes

5. Review Comments to the Author

Reviewer #1: The manuscript by Van Ginkel and collaborators is interesting. However, there are some points that need to be addressed by the authors.

1. The manuscript is not easily readable for not experts in the field. For example, the authors should clearly indicate what is and the mechanism of action of protamine sulfate. In the introduction should be added a brief explanation of how GAG and in particular hyaluronan is synthesized explaining the role of HASes. TEER technique should be briefly explained.

2. The authors should measure or localize TSG6 which is a HA binding protein with could have a role in the barrier function modifying HA.

3. The authors should show controls for the experiments with hyaluronidase and PS indicating a remodeling of the ECM.

4. Authors should check abbreviations throughout the text (i.e., sometimes they used HA sometimes hyaluronic acid...).

5. What is the molecular mass of commercial HA the authors added to digested samples (GAG-therapies)? What about if a different MW HA was used?

6. 4-methylumbelliferone, an aspecific inhibitor of HA synthases, could be used to show the role of HA in the barrier.

Reviewer #2: This is a very interesting research and article on function and location of HA in the urothelial cell.

A few comments for the authors:

Introduction: the authors should emphasize more on the novelty of this research, what will this research add to the existing publications on this topic.

Methods: the authors should add a justification of the use of porcine model for this research, since they used a human bladder tissue as well. What was the aim of the comparison between the human and porcine tissue?

Discussion: Is there any explanation as to why enzymatic digestion of HA in cell cultures did not lead to an increased permeability, moreover resulted in a more impermeable barrier? Why is this different with the removal of CS?

Another interesting phenomenon is that damage did not alter the expression of genes associated with barrier

function, is there any explanation which could be added in the discussion?

Reviewer #3: In this manuscript, van Ginkel at al show the localization of hyaluronic acid in tissue specimen of normal human and porcine urothelium, as well as evaluate different GAG replenishment therapies, used in clinical practice for treating interstitial cystitis, in in vitro model. Though the manuscript presents novel information and is potentially of interest, it presents with some flaws and it might lead to premature conclusions. Some additional experiments are required as well as improvements in the manuscript before it would be suitable for publication.

1. The third objective was to evaluate GAG therapies on an inflamed in vitro model, however, treating cells with PS does not lead to a substantial inflammation as seen in patients with IC/BPS. An in vitro model that employes PS together with another agent, such as LPS, TNFa or H2O2 should be used to better explain the effects of GAG therapies. I would strongly encourage the authors to add these additional experiments.

2. Methods: in all descriptions the number of biological replicates and technical replicates (number of independent experiments) is missing. Please add this information. Please also add how many human and porcine biopsies were used.

3. Why was the in vitro cell culture not prepared from human tissue since it was available?

4. Why did you decide to use different differentiation protocols for different experiments? If a very high TEER was achieved after 3 weeks of culture, why these cells were not used in subsequent experiments?

5. Why did you only evaluate gene expression of tight junctions? These should also be evaluated using ICH or IF, since gene expression does not always reflect the protein expression nor the distribution of tight junctions.

6. Statistical analysis: which post-hoc test was used to assess the differences between groups? Please add.

7. You state that HA was most abundantly present in the basal membrane of the urothelium, however, this should be labeled to justify your findings.

8. In figures with ICH, the scale bars are missing.

9. All figures should include information on how many biological/technical replicates, the labeling or other analysis was performed.

10. All figures should include an explanation of what we see: is it mean or median+/- SD, SEM?

11. Figure 3a: there is no legend explaining what individual lines represent.

12. Figure 4: please add what does the dotted line at y=1 indicate. Please also change graphs to show each individual value of a biological/technical replicate. Were there no significant results in this graph?

13. Why would GAG therapy increase inflammatory markers? I am not sure that the explanation about acute phase response is enough. If this is correct, then the expression of other markers that indicate resolution of inflammation or tissue fibrosis should also be evaluated.

14. If the integrity of urothelium recovers anyway after 24h then what is the point of treatment with GAG replenishment therapy? As written before, additional experiments with a more prominent inducement of inflammation should be employed.

15. For in vitro differentiation of NPU cells achieving high TEER values after 3 weeks, and resemblance with normal human urothelium, the protocol has already been described in literature. Please add the following citations:

https://link.springer.com/article/10.1007/s00418-014-1265-3

https://www.sciencedirect.com/science/article/pii/S0887233317302242?via%3Dihub

16. Why would be beneficial to generate a stronger urothelial barrier in healthy urothelium? But then in damaged urothelium, this therapy fails to improve the barrier? You are trying to replicate IC/BPS in which there is a leaky urothelium, do you implicate that this kind of therapy should not be used in IC/BPS? How would your findings translate into clinical settings?

17. Why would a mild inflammatory response increase impermeability of urothelium? In IC/BPS, there is an inflammatory response with increased permeability – inflammatory mediators are thought to further increase the damage of urothelial cells and further increase permeability though a positive feedback mechanism? How do you comment on this?

Reviewer #4: Presented manuscript deals with the function and occurrence of hyaluronic acid in urinary bladder epithelia. The manuscript is well written and uses appropriate methods to validate its claims. However, there are some minor issues that need to be resolved

1. Materials and methods, line 135 - The term »binocular microscopy« does not indicate the type of analysis. Bright field microscopy would be more appropriate in this case.

2. Results, line 193 – It is unusual to use the term basal membrane of the urothelium because in this multilayered epithelium each cell has its own basal part of the membrane. Since the hyaluronic acid is most likely located extracellularly its location can be described as occurring at the basement membrane or basal lamina. Fig. 2A.2 shows the situation after hialuronidase treatment, so that no labelling is visible. Presumably Fig. 2A.3 is supposed to show the porcine biopsy.

3. In Material and Methods section it is not described whether GAGs and hyaluronidase were added from the apical or basal side of the inserts. Since most proteoglicans and GAGs are located at the basal side, application from the apical side may not be effective if the epithelium is undamaged. For this reason, the application procedure should be described in the Material and Methods section.

4. Results, lines 267, 268 – Since treatment with low and high molecular mass GAG can give different results on the expression of genes for GAG or IL synthesis, the molecular mass of HA used in these experiments should be reported here?

5. Discussion, line 313 From the provided immages it is not possible to determine if HA is attached to the memberanes or even if this HA is located extracellularly or inside the cells. Aditionally, authors should explain what membrane-bound HA production means? HA is secreted from the cells by exocytosis and can only be attached to the membranes afterwards.

6. Discussion, lines 334,335 – Since inflammation usually decreases barrier function, the authors should explain how it could increase the impermeability of the urothelium?

7. Discussion, line 349 – The authors should explain why the addition of GAGs would increase the synthesis of new GAGs. Normally, the oversupply of molecules in the tissue decreases the synthesis of these molecules. The explanation should be confirmed by relevant literature.

6. PLOS authors have the option to publish the peer review history of their article (what does this mean? ). If published, this will include your full peer review and any attached files.

**Do you want your identity to be public for this peer review?** For information about this choice, including consent withdrawal, please see our Privacy Policy .

Reviewer #1: No

Reviewer #2: No

Reviewer #3: No

Reviewer #4: No

---

## [Author Response · Author response to Decision Letter 1]

20 Dec 2024

Reviewer #1

The manuscript by Van Ginkel and collaborators is interesting. However, there are some points that need to be addressed by the authors.

1. The manuscript is not easily readable for not experts in the field. For example, the authors should clearly indicate what is and the mechanism of action of protamine sulfate. In the introduction should be added a brief explanation of how GAG and in particular hyaluronan is synthesized explaining the role of HASes. TEER technique should be briefly explained.

Thank you for this valid comment. We have added more information to explain more the roles and mechanism of actions, to increase the readability.

2. The authors should measure or localize TSG6 which is a HA binding protein with could have a role in the barrier function modifying HA.

Thank you for this interesting contribution. There is indeed a lot of research showing an interaction between TSG-6 and HA. There is little known on the role of TSG-6 in IC/BPS specifically.

Previous research suggests that the function of TSG-6 (partly) relays on immune response/ immune cells with an interaction with CD44. We did not investigate the role of TSG-6 in the bladder. However, this is a very interesting possibility for future in vivo research or animal studies. We added this to the discussion section

3. The authors should show controls for the experiments with hyaluronidase and PS indicating a remodeling of the ECM.

We compared all experiments with untreated controls in both TEER and qPCR experiments. This information is also in the figures (qPCR up or downregulation compared to controls). A functional assay like TEER is matced with controls and gives a read out on epithelial barrier function realtiume. In our model TEER and therefore functionality is restored within 24 hrs, therefore ECM remodeling via other analyses such as SEM may provide some extra data, but we consider the current analyses sufficient to support the conclusions of this manuscript.

4. Authors should check abbreviations throughout the text (i.e., sometimes they used HA sometimes hyaluronic acid...).

Thank you, this inconsistency has been corrected.

5. What is the molecular mass of commercial HA the authors added to digested samples (GAG-therapies)? What about if a different MW HA was used?

The range for HA in Ialuril is between 1400KDalton and 2000KDalton. For Instylan the molecular weight is 2000KDalton. The MW has been added to the manuscript 85-88.

A higher molecular weight has been investigated, as so called superGag (crosslinked GAG molecules with higher molecular weights). For both CS and HA, these SuperGAGs have demonstrated an increased effectiveness in restoring barrier function in vitro, characterized by impermeability, compared to regular GAGs [2, 3]. So interestingly, this could have potential in future in vivo/clinical studies.

6. 4-methylumbelliferone, an aspecific inhibitor of HA synthases, could be used to show the role of HA in the barrier.

Indeed, there are additional modulating molecules (promotors/inhibitors) that could be used to further clarify the in vivo role of HA.

In the current study we chose to specifically remove HA (using hyaluronidase) and add HA (using GAG-therapy) artificially, in cell cultures. This gives the first insight in the role of HA in the urothelial barrier. To gain further insight in the complex dynamics of modulating molecules, that effect HA expression and function, additional research is necessary. Although this was outside of the scope of the current study, it would be a very interesting starting point future research.

Reviewer #2

This is a very interesting research and article on function and location of HA in the urothelial cell.

A few comments for the authors:

Introduction: the authors should emphasize more on the novelty of this research, what will this research add to the existing publications on this topic.

Thank you for this comment. We have rephrased parts of the introduction to better emphasizes the novelty of our research.

Methods: the authors should add a justification of the use of porcine model for this research, since they used a human bladder tissue as well. What was the aim of the comparison between the human and porcine tissue?

It is currently not feasible to obtain sufficient primary human urothelial cells from biopsies to perform these experiments. In our laboratory, efforts are underway to develop three-dimensional human cell cultures (organoids) for use in in vitro studies. Porcine bladder offers a matching anatomic and histochemical mimicry to the human bladder and therefore providing a practical alternative to facilitate these experiments. We used human and porcine bladder biopsies in our IHC experiments to show that there is this biologic mimicry and to show that our differentiated primary porcine cell cultures maintain there biological and histochemical properties, all in order to validate our in vitro model. By demonstrating similar characteristics between human and porcine tissue, we believe these studies yield valuable insights into the function of GAGs and GAG therapy in human tissue.

Discussion: Is there any explanation as to why enzymatic digestion of HA in cell cultures did not lead to an increased permeability, moreover resulted in a more impermeable barrier? Why is this different with the removal of CS?

Another interesting phenomenon is that damage did not alter the expression of genes associated with barrier function, is there any explanation which could be added in the discussion?

Earlier research from our group has shown that CS digestion leads to an increased permeability. Moreover, these results also showed that removal of Heparan Sulfate (HS) did not induce any alterations in the barrier function [4, 5]. Logically , we did not expect the results we obtained from our hyaluronidase experiments and amended our discussion to elaborate this response to HA digestion further. Taking this together with the results from our study, when considering the immunohistochemical staining of CS, HS and HA, it appears that the luminal presence of CS directly influences the barrier integrity. Conversely, GAGs found deeper within the urothelium, such as HS and HA, do not seem to directly compromise the barrier following digestion.

This has been added in 330-333

Regarding the lack of changes in barrier-related gene expression, we address this in the manuscript lines 414-415. Future research could explore broader gene analysis, other types of damage (e.g., TNF-α), or varying PS dose and duration.

Reviewer #3

In this manuscript, van Ginkel at al show the localization of hyaluronic acid in tissue specimen of normal human and porcine urothelium, as well as evaluate different GAG replenishment therapies, used in clinical practice for treating interstitial cystitis, in in vitro model. Though the manuscript presents novel information and is potentially of interest, it presents with some flaws and it might lead to premature conclusions. Some additional experiments are required as well as improvements in the manuscript before it would be suitable for publication.

1. The third objective was to evaluate GAG therapies on an inflamed in vitro model, however, treating cells with PS does not lead to a substantial inflammation as seen in patients with IC/BPS. An in vitro model that employes PS together with another agent, such as LPS, TNFa or H2O2 should be used to better explain the effects of GAG therapies. I would strongly encourage the authors to add these additional experiments.

Thank you for your thorough feedback. We used protamine sulphate in our experiment because of its very specific effect in neutralising the GAG-layer and increasing permeability, which was our primary objective for this study. PS is commonly used in in vitro models to increase the permeability of the urothelial barrier [6]. In our experiments it had the desired effect of a consistent response in increasing the permeability by inducing mild inflammation (reducing the TEER), without killing cells which is much more common in H2O2.

We have previously used LPS in non-published experiments, but results were less consistent in damage and recovery compared to protamine. Also, because this inflammatory agent is more commonly associated with bacterial infections and less with IC/BPS, we opted not to use it. TNF-alfa has indeed been used in IC/BPS research, and seems to create a good inflammation model, and is of interest due to its immune modelling effect. It would be very interesting to compare TNF-alfa with the protamine experiments for future research.

2. Methods: in all descriptions the number of biological replicates and technical replicates (number of independent experiments) is missing. Please add this information. Please also add how many human and porcine biopsies were used.

Good remark, this information has been added. (s.125, s.127, s.157, s164, s177)

3. Why was the in vitro cell culture not prepared from human tissue since it was available?

It is currently not feasible to obtain sufficient primary human urothelial cells from biopsies to perform these experiments and terminally differentiated urothelial cells with a very high barrier function were needed for our experiments. Therefore, human cell lines were not a good options and based on literature, immortalized cell cultures (so far) did not yield high TEER values compared to our in vitro model. In our laboratory, efforts are underway to develop three-dimensional human cell cultures (organoids) for use in in vitro studies. Porcine urothelial cells provide a practical alternative to facilitate these experiments. By demonstrating similar characteristics between human and porcine tissue, we believe these studies yield valuable insights into the function of GAGs and GAG therapy in human tissue. Ultimately, our goal is to culture human IC/BPS cells and conduct experiments directly on these human-based models.

4. Why did you decide to use different differentiation protocols for different experiments? If a very high TEER was achieved after 3 weeks of culture, why these cells were not used in subsequent experiments?

The cells for TEER measurements and for gene expression were cultured in different wells/membranes of different sizes. This was inherent to the materials needed for both experiments. We did not base the culture and differentiation protocol on the timeline but on evaluation of the cell growth and terminal differentiation.

Cells in the wells were grown until confluent and then differentiated for one week, this is indeed shorter than for the TEER experiment. We have experimented (unpublished ) with TEER measured in smaller and larger surface transwells and found that confluency plays a mayor part in differentiation. After one week of differentiation the cells were healthy and IHC showed a network of tight junctions indicating a differentiated (tight) epithelium(figure 1 B, ZO-1 expression).

TEER is measured in real-time experiment along time.

5. Why did you only evaluate gene expression of tight junctions? These should also be evaluated using ICH or IF, since gene expression does not always reflect the protein expression nor the distribution of tight junctions.

We did demonstrate tight junction formation in our in vitro model in figure 1. TEER function is highly dependent on tight junction formation, therefore we considered this as the best functional / dynamic outcome measure for complete barrier and tight junction function. IHC as outcome measure after damage could have provided some extra information on this, but is not conclusive of barrier function as mentioned in the discussion: ‘Corresponding with the observations of Turner et al., NPU cultures exhibited tight junctions and umbrella cells, even before the differentiation protocol was fully implemented. Tight junctions and umbrella cells, seen in histological IHC analyses, indicate the presence of differentiated cells and barrier markers, and urothelial tight junction impairment has been observed after protamine sulfate exposure. Our results show that this does not confirm impermeability, since these conditions still generated low TEER values (approximately <200 Ω∙cm²) indicating a leaky epithelium [8, 9].’

Tight junction IF analyses was therefore regarded as being an inferior outcome measurement to the real-time TEER measurements. Nonetheless; the effect of PS on tight junctions quality on IHF/IF has been shown by a our group in this previous study [7].

6. Statistical analysis: which post-hoc test was used to assess the differences between groups? Please add.

A Bonferroni post-hoc analysis was used. We have added this to the manuscript under ‘statistical analyses.’

7. You state that HA was most abundantly present in the basal membrane of the urothelium, however, this should be labelled to justify your findings.

Arrows are present in figure 2 IHC in A1 and A3, it shows a darker colouring around where the basal membrane of the urothelium is located. We have slightly altered the wording in our results to better describe our findings.

Furthermore, in figure 2 IHC B1 and B2 we have added what the apical side of the cell culture is, this shows that HA expression is most pronounced in cell membranes and cells just below the most apical layer.

8. In figures with ICH, the scale bars are missing.

We have tried various lay outs for our images, we eventually chose to show the magnification factor in the lower right corner of each image. We found this resulted in the most legible and least crowded overall image, while still showing al the necessary information.

9. All figures should include information on how many biological/technical replicates, the labeling or other analysis was performed.

We have added the replicates numbers to the figure captions.

10. All figures should include an explanation of what we see: is it mean or median+/- SD, SEM?

Thank you for the feedback, numbers a presented as mean +/- standard deviation. Where applicable, it has been added to the figure description text.

11. Figure 3a: there is no legend explaining what individual lines represent.

All different shades of blue/grey represent (individual) untreated insert cultures that were followed over time. This figure aims to show that all untreated cultured inserts increased in TEER over time.

A legend has been added.

12. Figure 4: please add what does the dotted line at y=1 indicate. Please also change graphs to show each individual value of a biological/technical replicate. Were there no significant results in this graph?

Expressions of hyaluronidase and protamine in figure 4 are relative and compared to the control/untreated of that respective timepoint. The expression in untreated T5-T24 are relative to untreated T3, the dotted line at y1 is this control line. This has been added to the text description figure 4. We have also added this to figure 5, the dotted line is also presented there.

The main goal of the figure is to show differences in relative expressions and the curves over time. Therefore, and to aid legibility, we did not add the exact numbers to the graphs. No statistical analysis was conducted due to the limited sample size. As technical replicates cannot be combined to account for variability, statistical testing was not feasible and therefore we reported descriptive results.

13. Why would GAG therapy increase inflammatory markers? I am not sure that the explanation about acute phase response is enough. If this is correct, then the expression of other markers that indicate resolution of inflammation or tissue fibrosis should also be evaluated.

This finding is ‘unexpected’ for GAG therapy. Most studies that evaluate GAG therapy do not evaluate inflammation in the very short time frame and this response is therefore likely missed in these studies. Inflammation responses and subsequent repair responses may differ in timing when it occurs and stops. Interestingly, if you look at a study by Rooney et al, where TNF-a-induced inflammation was treated with HA for 24 hours. They also observed a modest inflammatory reaction with GAG therapies with a modest initial increase in IL-8, similar to our findings. They found the decreasing effect of HA on IL-8

---

## [Decision Letter · Decision Letter 1]

14 Jan 2025

Hyaluronic acid: function and location in the urothelial barrier for interstitial cystitis/ bladder pain syndrome, an in vitro study

PONE-D-24-21268R1

Dear Dr. Ginkel,

We’re pleased to inform you that your manuscript has been judged scientifically suitable for publication and will be formally accepted for publication once it meets all outstanding technical requirements.

Kind regards,

Yung-Hsiang Chen, Ph.D.

Academic Editor

PLOS ONE

Additional Editor Comments (optional):

Congratulations on the acceptance of your manuscript, and thank you for your interest in submitting your work to PLOS ONE.

Reviewers' comments:

Reviewer's Responses to Questions

**Comments to the Author**

1. If the authors have adequately addressed your comments raised in a previous round of review and you feel that this manuscript is now acceptable for publication, you may indicate that here to bypass the “Comments to the Author” section, enter your conflict of interest statement in the “Confidential to Editor” section, and submit your "Accept" recommendation.

Reviewer #1: All comments have been addressed

Reviewer #3: All comments have been addressed

Reviewer #4: (No Response)

2. Is the manuscript technically sound, and do the data support the conclusions?

Reviewer #1: Yes

Reviewer #3: Yes

Reviewer #4: Yes

3. Has the statistical analysis been performed appropriately and rigorously? 

Reviewer #1: Yes

Reviewer #3: Yes

Reviewer #4: I Don't Know

4. Have the authors made all data underlying the findings in their manuscript fully available?

Reviewer #1: Yes

Reviewer #3: Yes

Reviewer #4: Yes

5. Is the manuscript presented in an intelligible fashion and written in standard English?

Reviewer #1: Yes

Reviewer #3: Yes

Reviewer #4: Yes

6. Review Comments to the Author

Reviewer #1: In this updated version of the manuscript, the Authors properly replied to all my concerns raised during the first round of revision.

Reviewer #3: (No Response)

Reviewer #4: The authors appropriately corrected all the outlined issues thus the manuscript is in my opinion suitable for further

procedure.

7. PLOS authors have the option to publish the peer review history of their article (what does this mean? ). If published, this will include your full peer review and any attached files.

**Do you want your identity to be public for this peer review?** For information about this choice, including consent withdrawal, please see our Privacy Policy .

Reviewer #1: No

Reviewer #3: No

Reviewer #4: No

---

## [Editor Report · Acceptance letter]

PONE-D-24-21268R1

PLOS ONE

Dear Dr. van Ginkel,

I'm pleased to inform you that your manuscript has been deemed suitable for publication in PLOS ONE. Congratulations! Your manuscript is now being handed over to our production team.

Kind regards,

on behalf of

Dr. Yung-Hsiang Chen

Academic Editor

PLOS ONE